



# Ablative and geomorphic effects of a supraglacial lake drainage and outburst event, Nepal Himalaya

Evan S. Miles[1], C. Scott Watson[1,2], Fanny Brun[3,4], Etienne Berthier[3], Michel Esteves[4], Duncan J. Quincey[1], Katie E. Miles[5], and Patrick Wagnon[4]

[1]School of Geography, University of Leeds, Leeds, LS2 9JT, UK
[2]Department of Hydrology and Atmospheric Sciences, University of Arizona, Tucson, AZ 85721, USA
[3]LEGOS, Université de Toulouse, CNES, CNRS, IRD, UPS, F-31400 Toulouse, France
[4]Univ. Grenoble Alpes, CNRS, IRD, Grenoble INP, IGE, F-38000 Grenoble, France
[5]Department of Geography and Earth Sciences, Aberystwyth University, SY23 3DB, Aberystwyth, UK

**Correspondence:** Evan S. Miles (e.s.miles@leeds.ac.uk)

**Abstract.** A set of supraglacial ponds rapidly filled between April and July 2017 on Changri Shar Glacier in the Everest region of Nepal, coalescing into a ~180,000 m$^2$ lake before sudden and complete drainage through Changri Shar and Khumbu Glaciers 15-17 July. We use a suite of PlanetScope and Pléiades satellite orthoimagery to document the system's evolution over its very short filling period and to assess the glacial and proglacial effects of the outburst flood. We additionally use high resolution

stereo digital elevation models (DEMs) to complete a detailed analysis of the event's ablative and geomorphic effects. Finally, measurement of the flood's passage at a stream gauge 4 km downstream enables a refined interpretation of the chronology and overall magnitude of the outburst. We infer largely subsurface drainage through both glaciers located on its flowpath, and efficent drainage through Khumbu Glacier. The drainage and subsequent outburst of $1.36\pm0.19\times10^6$ m$^3$ impounded water had a clear geomorphic impact on glacial and proglacial topography at least as far as 11 km downstream, including deep incision

and landsliding along the Changri Nup proglacial stream, the collapse of shallow englacial conduits near the Khumbu terminus and extensive, enhanced bank erosion below Khumbu Glacier. These sudden changes led to the rerouting of major trails in three locations, demonstrating the potential hazard that short-lived, relatively small glacial lakes pose.

## 1    Introduction

Outburst floods occur due to the sudden release of stored water from glaciers, which can be stored at the glacier surface

within topographic lows (Benn et al., 2012; Chu, 2014); internally along englacial conduits, crevasses, and voids (Fountain and Walder, 1998); and at the glacier's bed (Jansson et al., 2003). Water can also be impounded by the glacier or its moraines to form ice-marginal or proglacial lakes; outburst floods from such lakes can lead to catastrophic geomorphic change and subsequent societal impacts reaching far downstream, and have been a topic of focused study in High Mountain Asia (e.g. Benn et al., 2012; Westoby et al., 2014; Rounce et al., 2016; Narama et al., 2018; Nie et al., 2018; Veh et al., 2018) and

globally (e.g. Carrivick and Tweed, 2016; Cook et al., 2016; Harrison et al., 2018). Outburst floods from water stored within the glacier system are generally smaller in magnitude, but they can occur repeatedly due to seasonal and interannual variations





in a glacier's hydrological system, whether impounded supraglacially and englacially (e.g. Benn et al., 2017; Miles et al., 2017b; Rounce et al., 2017; Narama et al., 2017; Watson et al., 2017) or subglacially (e.g. Walder and Driedger, 1995; Wadham et al., 2001; Garambois et al., 2016), or due to ice-marginal dynamics (Huss et al., 2007; Steiner et al., 2018). These storage components are interlinked: water retained at the surface can reach englacial and subglacial systems through hydrofracture or

exploitation of zones of permeability (e.g. Gulley et al., 2009), while water impounded within or beneath the glacier can drain surficially if water pressures rise substantially (e.g. Roberts et al., 2002).

Despite their smaller magnitude, glacier outburst floods that emanate from supraglacial and englacial sources can be severely damaging to infrastructure, yet they have not received focused study in the Himalaya (Richardson and Quincey, 2009; Rounce et al., 2017). The low density of hydrologic gauging stations limits hydrograph observation, while aerial and satellite observa-

tion of supraglacial water storage is hampered by the South Asian Monsoon, obscuring the glacier surfaces with clouds when supraglacial ponding is most prevalent (Watson et al., 2016; Miles et al., 2017b). Nonetheless, recent observations have indicated that these smaller floods can occur with regularity and have the potential to be hazardous (Rounce et al., 2017; Narama et al., 2018).

Changri Shar Glacier is a valley glacier in the Everest region of Nepal (Figure 1). The glacier is characterised by a 4.0

km$^2$ debris-covered tongue extending from an elevation of ~5500 m a.s.l to the terminus at ~5070 m a.s.l. The thick surface debris of the glacier tongue greatly retards surface ablation and leads to hummocky surface topography. Changri Shar and the neighbouring Changri Nup Glacier (Vincent et al., 2016; Sherpa et al., 2017) discharge water into a proglacial gorge, which funnels water into the true-right side of Khumbu Glacier, for which the glaciers constituted a former tributary. The stream has cut into the lateral margin of Khumbu Glacier, leading to development of a large bare ice cliff, and this water flows englacially

or subglacially to join the Khumbu drainage system. Changri Shar, Khumbu, and other debris-covered glaciers in the area are generally responding to local climate warming through surface lowering and stagnation, rather than retreat (e.g. Rowan et al., 2015; King et al., 2017). These factors combine to create very low surface gradients for the lower ablation area, and increase the likelihood of formation of large proglacial or supraglacial lakes in this zone (Quincey et al., 2007; Miles et al., 2017b; King et al., 2018).

In the pre-monsoon period of 2017, a large supraglacial lake developed over a period of three months on the Changri Shar Glacier, and drained suddenly within a short window in the monsoon. Here, we combine PlanetScope, RapidEye, and Pléiades optical satellite imagery along with field observations and a discharge record to document the expansion and drainage of this supraglacial lake system, and to describe its ablative and geomorphic impacts on the Khumbu Glacier, through which the flood travelled. Finally, we highlight the impact of the flood on the downstream river system by quantifying rates of bank erosion

and channel migration.





## 2 Methods

### 2.1 Supraglacial lake area

To document the supraglacial lake expansion, we analysed 25 Level 3B tiles collected by the PlanetScope Dove satellite constellation between 27 March 2017 and 26 October 2017 (Table S1). These 4-band data have a ground sampling distance of

3.7 m but are resampled to 3 m during orthorectification, and digital numbers (DNs) contain scaled at-sensor radiance values for the Blue (B: 455-515 nm), Green (G: 500-590 nm), Red (R: 590-670 nm), and Near-Infrared (NIR: 780-860 nm) spectral ranges. We additionally used several RapidEye level 3B tiles for pond coverages and geomorphic interpretations. These are 5-band data (B: 440-510 nm; G: 520-590 nm; R: 630-685 nm; Red Edge: 690-730 nm; NIR: 760-850 nm) with a ground sampling distance of 6.5 m, resampled to 5 m during orthorectification (Planet Team, 2017). Due to the high density of clouds during

the monsoon, few scenes are cloud-free over the full study area; we therefore manually masked clouds and cloud shadows in the region of the supraglacial lake before mapping ponded water (inset panels, Figure 1). For each scene, we calculated the Normalised Difference Water Index (NDWI) based on DNs for the G and NIR bands ($\text{NDWI} = \frac{G - NIR}{G + NIR}$; e.g. McFeeters, 1996) and used an Otsu adaptive histogram-based approach to select an optimised NDWI threshold (Otsu, 1979; Cooley et al., 2017), identifying ponded water as those pixels exceeding this threshold. Finally, the pond cover products were again manually

inspected for removal of terrain and cloud shadows before development of a lake area time series, and we use a $\pm$1-pixel buffer for lake area uncertainty (e.g. Gardelle et al., 2011).

### 2.2 DEM generation and surface elevation changes

We then analysed two along-track Pléiades triplets (Berthier et al., 2014) with acquisition dates of 23 March 2017 and 14 December 2017, bounding the lake's filling and drainage. The two scenes had maximum base-to-height ratios of 0.55 and

0.32, respectively. Their panchromatic bands (480–830 nm, ground sampling distance of 0.7 m) were processed using the Ames Stereo Pipeline (Shean et al., 2016) to generate DEMs and orthoimages at 2 and 0.5 m resolution, respectively. The two Pléiades DEMs were 3D-coregistered using off-glacier terrain (Berthier et al., 2007), then differenced to produce a map of surface elevation change (dH) spanning the 2017 monsoon period. This geodetic difference encompassed the majority of the ablation season, so for the glaciers we focused on zones of heightened surface lowering not solely attributable to ice cliffs

and supraglacial ponds, which are known hot spots of melt for Himalayan debris covered glaciers (Immerzeel et al., 2014; Thompson et al., 2016; Ragettli et al., 2016). We thus identified 11 zones of prominent elevation change that were clearly associated with the lake drainage according to the PlanetScope and RapidEye imagery to interpret the ablative and geomorphic effects of the supraglacial lake drainage (Table 1, Figure 2). Field visits in May 2017, October 2017, and May 2018 enabled direct observation of many of the most prominent zones of change.

To assess the error on the elevation difference obtained by differencing of two Pléiades DEMs, we follow the tile methods of Berthier et al. (2016) and split the stable terrain dH maps into $n \times n$ tiles, with $n$ varying from 2 to 200. The corresponding individual tile area thus varies from 91.2 km$^2$ ($n = 2$) to 0.04 km$^2$ ($n = 200$). For each tile, we compute the absolute value of the median dH. We then calculate our dH error ($\sigma_{dH}$) as the average of these $n^2$ absolute values, and $\sigma_{dH}$ ranges from 0.12



m ($n = 2$) to 0.64 m ($n = 200$). In the Figure 2 inset, dH is plotted as a function of the individual tile area. The relationship is well represented by a logarithmic fit which we use as our error model. Consequently for all our zones of change we estimate an error based on the zone area, and only analyse elevation changes of magnitude greater than this error.

## 2.3 Lake volume estimation

Using the pond-free March 2017 Pléiades DEM, we identified 142 surface depressions and determined area-volume relationships for each by progressively filling the surface depressions with an increment of 0.1 m depth (as in, e.g. Watson et al., 2017). We then calculated stored water volumes in the supraglacial lake area for each PlanetScope scene by estimating the volume of each individual pond in the area of the supraglacial lake, then summed these to estimate the total ponded volume in the study area (Figure 3). On 16 July the lake was partially obscured by cloud, so we instead estimated the water level and volume

from a partial shoreline dataset (Figures S1 and S2). This approach assumes very minor topographic changes in the proximity of the supraglacial lake during the study period, but many studies have noted the local ablative effects of supraglacial ponds (Benn et al., 2001; Röhl, 2008; Brun et al., 2016; Miles et al., 2016; Salerno et al., 2017). Thus, the resulting volume estimates carry considerable uncertainty (in this case calculated using the ±1-pixel areal uncertainties), but are nonetheless useful and conservative values of supraglacial water storage during this period.

## 15 2.4 Proglacial bank erosion and channel migration

We also measured areal changes associated with active channel migration and bank erosion along the Khumbu proglacial stream using RapidEye level 3B imagery from November of 2012, 2015, 2016, and 2017. The images were coregistered in ENVI (RMSE < 1 m), then we calculated changes in the NDWI and Normalised Difference Vegetation Index (NDVI = $\frac{NIR-R}{NIR+R}$) for 2012-2015 and 2016-2017, enabling us to resolve periods preceding and encompassing the 2017 outburst from Changri Shar.

We considered the major NDVI changes (all decreases) to indicate bank erosion and reactivation, while strong NDWI changes indicate stream migration. We calculated a 3x3 focal mean to reduce noise, then eliminated low-magnitude changes in the indices based on a visual inspection of the histogram (thresholds in Table 2). We manually trimmed the results to zones within the channel, also eliminating areas severely affected by shadows. Finally, we aggregated areas of bank erosion and stream migration in 1 km bins along the main Khumbu Khola to compare rates of change preceding and bounding the event (Table 2).

## 25 2.5 Discharge measurements at Pheriche

Finally, the study period coincided with automated water level measurements collected every 30 minutes in the proglacial stream near Pheriche village (Figure 1). A rating curve has been developed for this position based on 34 field-calibrated fluorescein discharge measurements collected since November 2010 and was used to calculate discharge for the period of analysis. Based on the analyses of Di Baldassarre and Montanari (2009) and McMillan et al. (2012), we estimated a discharge uncertainty

of 15% for stage values within the calibrated range and 20% for stage values above the maximum stage-discharge measurement. From this record, we estimate normal background discharge (hereafter, baseflow) for 17:00 on 15 July to 09:00 on 17



July (all times given in Nepal Time, NPT; UTC +05:45) using a half-hourly cubic spline interpolant fitted to measurements for 10-15 and 17-20 July (i.e. interpolating between preceding and subsequent 09:30 measurements to estimate discharge at 09:30 on 16 July), and determine the flood discharge as the difference between observed discharge and estimated baseflow.

## 3 Results

Prior to 2017, the area of the supraglacial lake was characterised by occasional ponds filling and draining, both seasonally and interannually. Surface depressions in the study area began to accumulate water in March 2017 (Figure 3), likely due to the seasonal blockage of shallow subsurface englacial pathways (Benn et al., 2017; Irvine-Fynn et al., 2017; Miles et al., 2017a). The isolated ponds grew and coalesced rapidly to encompass an area of 160,000±15,400 m$^2$ during 7-13 July (26% of the area inset in Figure 1); based on our topographic analyses, we estimate a lake volume of $1.36 \pm 0.19 \times 10^6$ m$^3$ for this date. Drainage had begun by 16 July, when we estimate that the lake system's area and volume had reduced to 75,600±11,100 m$^2$ and $0.35 \pm 0.034 \times 10^6$ m$^3$ (this estimate is based on limited shoreline data; Supplementary Material). The lake's area had stabilised by 17 July, leaving several isolated ponds containing 44,000±15,000 m$^3$, which changed little thereafter in 2017 (Figure 1h-i).

Visual inspection of the Planet optical imagery and Pléiades DEMs reveals little change in the area immediately down-glacier of the lake following drainage. Near the terminus of Changri Shar, pronounced surface lowering was concentrated along the proglacial/supraglacial stream (Zone A in Figure 2 and Table 1). Where this stream leaves the glacier system, it destabilised the northern side of Changri Shar's proglacial gorge (Figure 4), leading to a large landslide by 16 July (Zone B). The erosion in this area forced reestablishment of a major trail between Lobuche and Gorak Shep settlements on the trek to Everest Base Camp.

On 16 July, the Changri Shar proglacial stream entry point into Khumbu Glacier was clearly observed to be buried by the mixed water and debris slurry from the initial outburst flood and the Zone B landslide. Based on the observed area of the inundated zone (32,700 m$^2$) and the March Pléiades DEM, we estimate a total volume of $2.56 \times 10^5$ m$^3$ impounded at the Khumbu entry on 16 July (Figure 4). By 17 July, the Changri Shar stream had incised through the debris deposit, and large concentric crevasses had opened in Khumbu Glacier surrounding this point; field observations confirmed that these features are still apparent in 2018. This area experienced a mean surface lowering of 6 m for the March-November period, totalling a volume loss of $1.86 \times 10^5$ m$^3$ despite the significant debris deposition, of which at least 32,900 m$^3$ remained in December (Zones C and D in Table 1).

There is little evidence of surface change on Khumbu Glacier relating to the drainage event until a point 2.8 km down-glacier from this entry point. Here, some 2.3 km upstream of the Khumbu terminus, large zones of pronounced surface lowering and supraglacial channel migration are apparent in the dH map and 16-17 July orthoimages, and cannot be accounted for by pre-existing ice cliffs (Figure 5). We interpret these to be collapse features following the route of shallow englacial channels which were exploited by the floodwaters (Zones E-H). These zones of enhanced surface change continued to the Khumbu Glacier terminus and account for at least $4.53 \times 10^5$ m$^3$ of volume loss (Table 1). Field observations of the lower ablation area in April





2018 suggested that additional conduits collapsed and became exposed at the surface in this area through winter ($>$ 9 months after the event), beyond the observation period of the March-November DEM difference.

The Khumbu Glacier proglacial stream system also underwent extensive changes during 2016-2017, including widespread patterns of stream migration and bank erosion (Figure 6). During this period, the stream destabilised the moraine outlet, leading to small landslides (Zones I and J). Directly below the Khumbu outlet the proglacial stream overflowed its banks, leading to areas of considerable erosion and deposition ($>$ 3 m dH) across the outwash plain (Zone K, Figure 6). Below the outwash plain, the proglacial channel showed patterns of active channel migration and bank erosion between 16-17 July and at least as far as Pangboche (11 km downstream), with analysis further down-valley inhibited by deep terrain shadows. The total area affected by channel migration (52,700 m$^2$) for the 2016-2017 period is similar to total channel migration over 2012-2015 (Table 2), but the 2016-2017 period exhibits a greatly magnified area of bank erosion (117,200 m$^2$ vs 6,125 m$^2$).

The proglacial river stage record near Pheriche documented seasonal and diurnal variations in discharge (Figure 7, inset). Discharge was $<$ 2 m$^3$ s$^{-1}$ prior to June 2017, then stabilised at ~3 m$^3$ s$^{-1}$ until the beginning of July. Early July was characterised by greater variation in discharge, with daily peaks up to 10 m$^3$ s$^{-1}$ declining into the middle of July. On 15 July, the discharge record departed from this general decline in peak daily flow, and discharge progressively increased to peak at $56 \pm 11$ m$^3$ s$^{-1}$ at 12:30 on 16 July. Discharge decreased rapidly after 13:00 to a low value of 5.9 m$^3$ s$^{-1}$ at 17:30, then again increased to 12.4 m$^3$ s$^{-1}$ at 20:30. Measured discharge then decreased gradually to 2.9 m$^3$ s$^{-1}$ at 10:00 on 17 July, and resumed a regular diurnal pattern with discharge varying between 3-7 m$^3$ s$^{-1}$. Based on our estimated baseflow, we calculated a total flood discharge of $0.97 \pm 0.23 \times 10^6$ m$^3$ between 20:00 on 15 July and 10:00 on 17 July.

## 4 Discussion

### 4.1 Interpretation

The dynamics of the lake system formation are relatively straightforward to interpret. A significant obstruction to the coupled supraglacial and englacial drainage system must have formed during winter 2016-2017, as occurs seasonally for other debris-covered glaciers (Benn et al., 2017; Miles et al., 2017a). This may have been the consequence of a significant conduit collapse or freeze-on of accumulated englacial debris, as has been observed through glaciospeleology (e.g. Gulley and Benn, 2007; Gulley et al., 2009), but the impediment to drainage was unusually effective in early 2017, preventing the development of preferential flowpaths which would lead to increasingly efficient drainage. Thus, as winter snow in the ablation area melted due to the onset of pre-monsoon conditions, this water accumulated in a large surface depression opened over recent years by heightened ablation along supraglacial ponds and ice cliffs. The accumulated water would have had a positive surface energy balance through the pre-monsoon, leading to peripheral ablation and further increasing the depression capacity and lake volume (Sakai et al., 2000; Benn et al., 2001; Miles et al., 2016).

The ponds initially grew in isolation, then coalesced supraglacially between 18 May and 17 June as the water levels rose (Figure 3). By 19 June, new peripheral ponds began to fill, suggesting the flooding of englacial conduits to a distance of 300 m from the main water body. These secondary ponds mostly coalesced with the main surface water body before its eventual





drainage. Based on the pond shorelines and Pléiades DEM, we estimate a steady water supply rate of 0.14 m$^3$ s$^{-1}$ for 17 June to 13 July.

The dynamics of pond drainage are slightly less clear due to the lack of observations during 14-15 July. Drainage of the lake began between 13-15 July, and was still underway on 16 July according to the PlanetScope imagery. Given the total duration
of the flood at Pheriche (~36 hours) and the landslide deposit on the 16th, we expect that drainage began around midday on 15 July. Based on the lack of down-glacier surface change on Changri Shar, the lake must have drained englacially or subglacially; this could have been accomplished by penetrating the internal blockage or via hydrofracture. In either case the water reemerged at the surface 700 m away, just prior to the Changri Shar terminus.

The textureless appearance of the flooded entrance to Khumbu Glacier imaged on 16 July (Figure 3c) suggests that the water
had only recently reached this position; this assessment is supported by the rapid subsequent drainage of the flooded water and incision of the debris deposit, which had occurred by 17 July. As this subsurface conduit would have closed at least partially due to creep since the prior monsoon, the sudden input of water and debris likely overwhelmed the conduit's capacity. Using an empirical relation for peak tunnel discharge ($Q_p = 46V_p{}^{0.66}$, with $V_p$ the lake volume in $10^6$ m$^3$; Walder and Costa, 1996), we estimate a peak discharge of 59 m$^3$ s$^{-1}$. Some water may have been retained in the glaciers' drainage network, and the
flood at Pheriche is likely to have incorporated additional meltwater and debris along its glacial and proglacial flowpath, but this discharge estimate is very close to the maximum discharge of $56 \pm 11$ m$^3$ s$^{-1}$ observed at the Pheriche gauge.

As with Changri Shar, the lack of surface change on Khumbu Glacier suggests a subsurface flowpath for much of the glacier's length. However, the floodwaters appear to have reached the glacier surface 2.3 km from the terminus, where several segments of conduit collapse are evident; this is in part due to the heightened hydrological base level of Khumbu Glacier, whose terminus
area has experienced extensive ponding in recent years (Watson et al., 2016).

The contrast in proglacial stream migration and bank erosion magnitudes between the 2012-2015 and 2016-2017 periods is clear (Table 2). As evidenced by the 2012-2015 period, channel migration is a continuous background process, but largely stays within the stream banks. This period encompasses the Gorkha earthquake (Kargel et al., 2016), which would have enhanced debris supply and stream migration. Outburst floods from Imja Khola during 2015 and 2016 (Rounce et al., 2016) may also
have affected landscape change below Pheriche. However, the 2016-2017 NDWI and NDVI changes show a greater magnitude of channel migration despite the shorter interval. The area of bank erosion is greatly magnified during 2016-2017, and examining available historic satellite image archives we have not found an image of this area in similar conditions, suggesting this magnitude of geomorphic change is uncommon.

The double peak of discharge observed at Pheriche is unusual for outburst floods. One possible cause is the blockage of the
Khumbu stream inlet by the landslide in the Changri Shar proglacial gorge (Figure 4). This is likely to have initiated around peak flow through the gorge, and would have led to a precipitous decline in discharge, followed by a later, sudden increase as preferential flowpaths developed through the debris (Gulley et al., 2009). Alternatively, it is possible that the heightened discharge late on 16 July corresponds to delivery of other water stored within the glacier system. Such stored water might connect to the drainage system more efficiently by the opening of conduits and channels during the flood. It is clear that the





heightened discharge at Pheriche only lasts until 10:00 on 17 July, so either mechanism had a short-lived influence on the glaciers' overall discharge.

## 4.2  Implications

The utility of novel observational platforms for observing and interpreting this event is noteworthy, and this supraglacial lake

drainage and subsequent outburst has several implications for cryospheric hazards and debris-covered glacier hydrology. First, this is an extremely short-lived event, with a lake system of $1.36 \times 10^6$ m$^3$ (544 Olympic swimming pools) filling and draining within one ablation season. This is important because despite the lake's short duration and relatively small volume, the event led to considerable glacial, fluvial, and geomorphic change, and forced diversions of major trails, the primary corridor for local trade and tourism, in at least three locations (Figures 2, 5 and 6; Watson and King, 2018). As suggested by Komori et al.

(2012) and Narama et al. (2018), the hazard posed by such features is non-negligible, yet traditional glacial lake monitoring approaches, which rely on repeat optical imagery such as Landsat and Sentinel-2, had difficulty observing the lake's formation at all due to the timing of repeat passes and cloud cover. Considering all Landsat 8 or Sentinel-2 scenes, we find only two that are mostly cloud-free over the supraglacial lake in the two months leading up to lake drainage. Pond observations during the monsoon are intermittent at best (Watson et al., 2016; Miles et al., 2017b) and thus we recommend the adoption of high-

frequency repeat optical imagery (as in this study) and Synthetic Aperture Radar data products (e.g. Strozzi et al., 2012) for improved monsoon monitoring of glacier hydrology.

Furthermore, the limited seasonal observations (biased to closely monitored glaciers) suggest that short-lived or seasonal outburst floods may be a regular feature for debris-covered glaciers in the region. This is important because in both the cases of Rounce et al. (2017) and this study, outburst floods from sources other than large proglacial lakes had downstream effects

on the transportation networks and livelihoods of local communities. The several observations of seasonal outburst floods are suggestive of a distinct seasonal cycle of hydrological development for debris-covered glaciers as compared to clean ice glaciers (e.g. Fyffe et al., 2015; Miles et al., 2017b; Narama et al., 2017). Rather than a gradual up-glacier progression of an efficient, connected drainage network (e.g. Nienow et al., 1998), these glaciers may impound significant volumes of water internally and at the surface before establishing efficient drainage through the lowest portion of the glacier. This key difference

is likely related to the melt-inhibiting thick debris low on such glaciers, which reduces the terminus area's sensitivity to seasonal warming. Instead, the zone of maximum melt (and seasonal sensitivity) is usually in the middle of the ablation area, leading to significant meltwater generation before efficient drainage pathways have been established for the lower glacier (Benn et al., 2017).

Nevertheless, the geomorphic evidence suggests that supraglacial lake outburst floods of this magnitude are not particularly

common (indeed, no large supraglacial lake is forming on Changri Shar in 2018, and past years show no evidence of such a lake). Still, supraglacial water storage is increasing for many Himalayan glaciers (e.g. Thompson et al., 2012; Watson et al., 2016). This is expected as climate warms and debris-covered glaciers stagnate, precursors to proglacial lake formation (Benn et al., 2012). In the case of Changri Shar, a very large closed surface depression had been opened by ice cliffs and supraglacial ponds prior to this event, creating the storage capacity for the $1.36 \times 10^6$ m$^3$ lake we observe. Consequently, as the excavation





and pitting of near-stagnant debris-covered glacier termini by ice cliffs and supraglacial ponds becomes more prevalent with a warming climate, other glaciers in the regional may accumulate large supraglacial water bodies. While the coalescence of ponds to form a large supraglacial lake represents an early stage of base-level lake development (Watanabe et al., 2009; Benn et al., 2012), such supraglacial lakes also outburst (as evidenced here). Thus, the expected increase in moraine-dammed glacial

lake outburst floods due to a lagged response to climate warming (Harrison et al., 2018) may also apply to the outburst of supraglacial water bodies, and events similar to the Changri Shar outburst may become more commonplace.

Finally, the rapid transit time we observe for the flood's passage of the lower Khumbu Glacier suggests that the glacier's subsurface drainage system can adapt an efficient configuration given sufficient water inputs. We base this assessment on the sudden interruption of peak discharge observed at Pheriche, which most likely corresponds to the blockage of the Changri

Shar stream portal as observed in the PlanetScope image on 16 July (Figure 4c). This image was captured at 09:50, implying a transport time of 3-5.5 hours for water to travel a straight-line distance of 4.9 km through Khumbu Glacier. Consequently we estimate a conservative mean travel velocity of 0.25-0.45 m s$^{-1}$; the water also passed 4 km from the glacier to Pheriche during this time. Prior dye-tracing studies have considered flow velocities $> 0.2$ m s$^{-1}$ to indicate hydraulically efficient drainage through a system of major conduits (e.g. Hubbard and Glasser, 2005), which we interpret to be the case for drainage

through Khumbu Glacier during this event. It is likely that subsurface drainage exploited a preexisting flowpath maintained by normal discharge from Changri Shar and Changri Nup Glaciers, as inferred for Ngozumpa Glacier by Benn et al. (2017), thus enabling the system's rapid adaptation to surplus water. It appears that subglacial or deep englacial flowpaths were utilised by the flood for both Changri Shar (~700 m subsurface transit) and Khumbu (2.8 km subsurface transit) glaciers, largely bypassing the coupled supraglacial-englacial drainage networks inferred by Irvine-Fynn et al. (2017) and Miles et al. (2017a).

**5   Conclusions**

We applied high resolution satellite remote sensing imagery to document and interpret the rapid formation, drainage, and outburst of a supraglacial lake system on Changri Shar Glacier in the Everest region of Nepal. The lake filled in ~3 months to encompass an area of 180,000 m$^2$ and volume of $1.36 \times 10^6$ m$^3$ prior to drainage, likely beginning on 15 July. The flood appears to have passed primarily through the subsurface of both Changri Shar and Khumbu glaciers. With a peak discharge

of 56±11 m$^3$ s$^{-1}$ observed 4 km downstream and glacier transport velocities of 0.25-0.45 m s$^{-1}$, the event is suggestive of an efficient subsurface drainage system configuration largely bypassing the coupled supraglacial-englacial systems common to hummocky debris-covered glaciers. The outburst flood led to substantial geomorphic change for both the Changri Shar and Khumbu proglacial systems, and forced rerouting of major trails in the area. We expect that outburst floods of this type and magnitude are not common, but may increase due to climate warming and consequent glacier recession.

Our observations of lake dynamics were only possible through the use of rapid-repeat high-resolution imagery, and similar approaches should be used to document monsoon-season hydrology of debris-covered glaciers, which is largely unobservable by traditional optical satellite sensors. There is evidence for dynamic changes to these glaciers' drainage systems during the monsoon and for seasonal outbursts of lower magnitude as a common feature. Nonetheless, there remains a considerable



need for systematic, robust observations of debris-covered glacier hydrology, as these glacier systems exhibit distinct storage components and seasonal drainage development patterns to clean ice glaciers. This is a crucial observational gap, as the hydrological storage and discharge of debris-covered glaciers has significant consequences for glacial hazards, surface ablation, glacier dynamics, proglacial sediment dynamics, and water supply with direct effects on downstream populations.

5 *Data availability.* All derivative data used in this study (lake coverages, dH zones) are available upon request. Please contact Evan Miles for this purpose (e.s.miles@leeds.ac.uk). PlanetScope and RapidEye data are freely available in reasonable quantities for research and education, see https://www.planet.com/markets/education-and-research/.



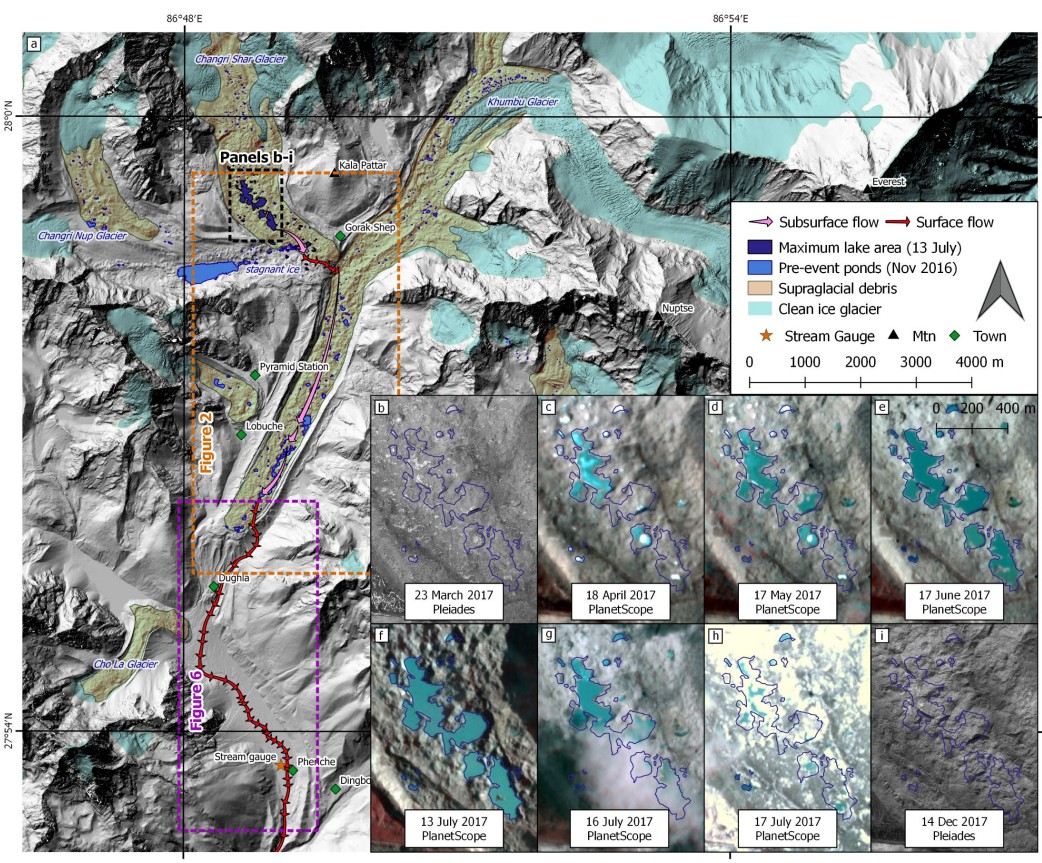

**Figure 1.** The study area and interpreted flow path through Changri Shar and Khumbu Glaciers, and the expansion and drainage of the Changri Shar supraglacial lake in 2017. Debris-covered glacier area was delineated manually with respect to the March Pléiades imagery, and modified from the RGI 6.0 (Pfeffer et al., 2014). The background hillshade is a composite from Pléiades (this study) and WorldView sources (Shean et al., 2016).



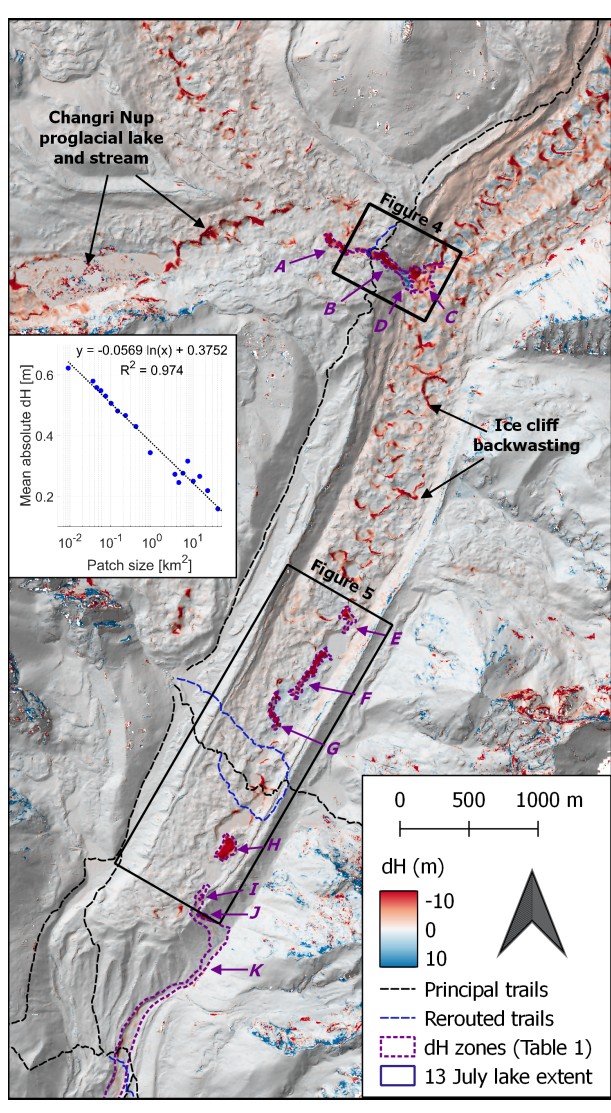

**Figure 2.** Zones A-K (purple labels) of ablative and geomorphic change associated with the lake drainage as measured by Pléiades March-December DEM differencing, with context of Figures 4 and 5 indicated.



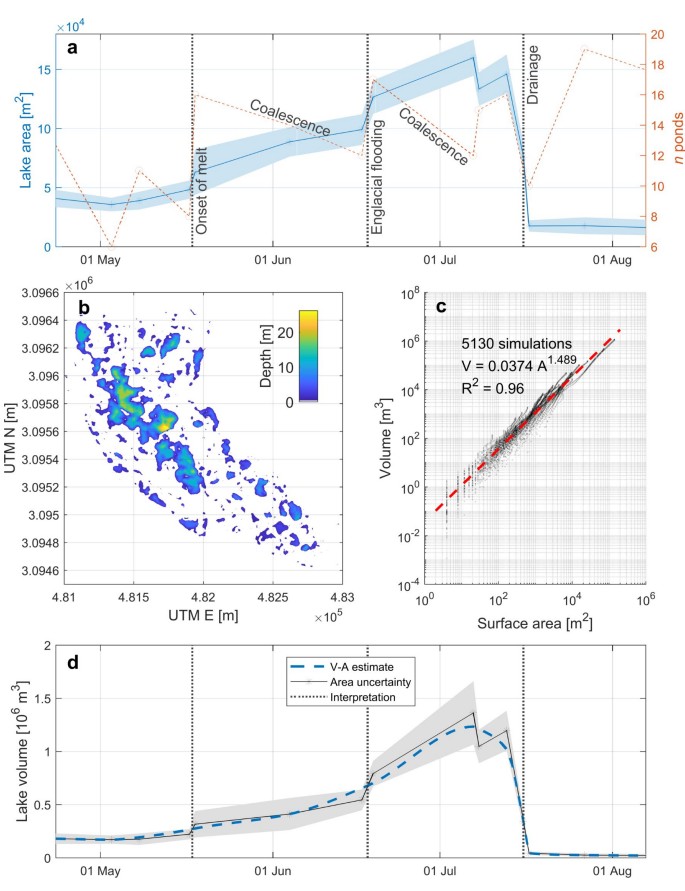

**Figure 3.** Total lake area and number of individual water bodies within the area of the insets in Figure 1 during the supraglacial lake's expansion and drainage, expressing areal uncertainty with a $\pm 1$-pixel buffer (a). An analysis of the depth of closed surface depressions on Changri Shar Glacier from the March 2017 Pléiades DEM (b) was used to determine the volume-area relationship for the study area (c). We used this relationship to reconstruct the lake system's volume prior to drainage (d).



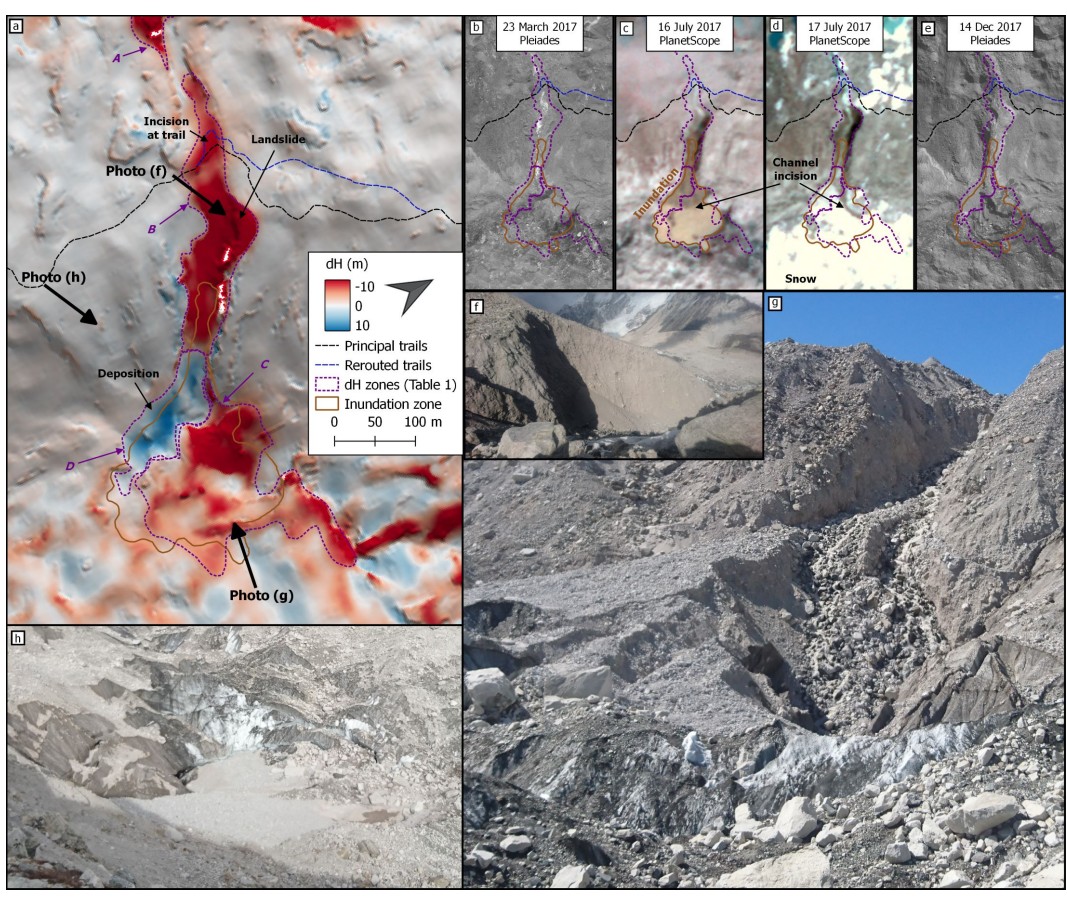

**Figure 4.** Surface elevation changes at the base of the Changri Shar proglacial gorge, also indicating locations for field photos (a). Panels (b-e) document the blockage and incision of the stream inlet to Khumbu Glacier. A fresh landslide scarp near the top of the proglacial gorge, the likely source for much of the debris (f), as viewed in May 2018. The deposit and incised channel as viewed from the Khumbu Glacier surface (g) in October 2017. The deposit and concentric crevassing as viewed from the Khumbu moraine in October 2017 (h).



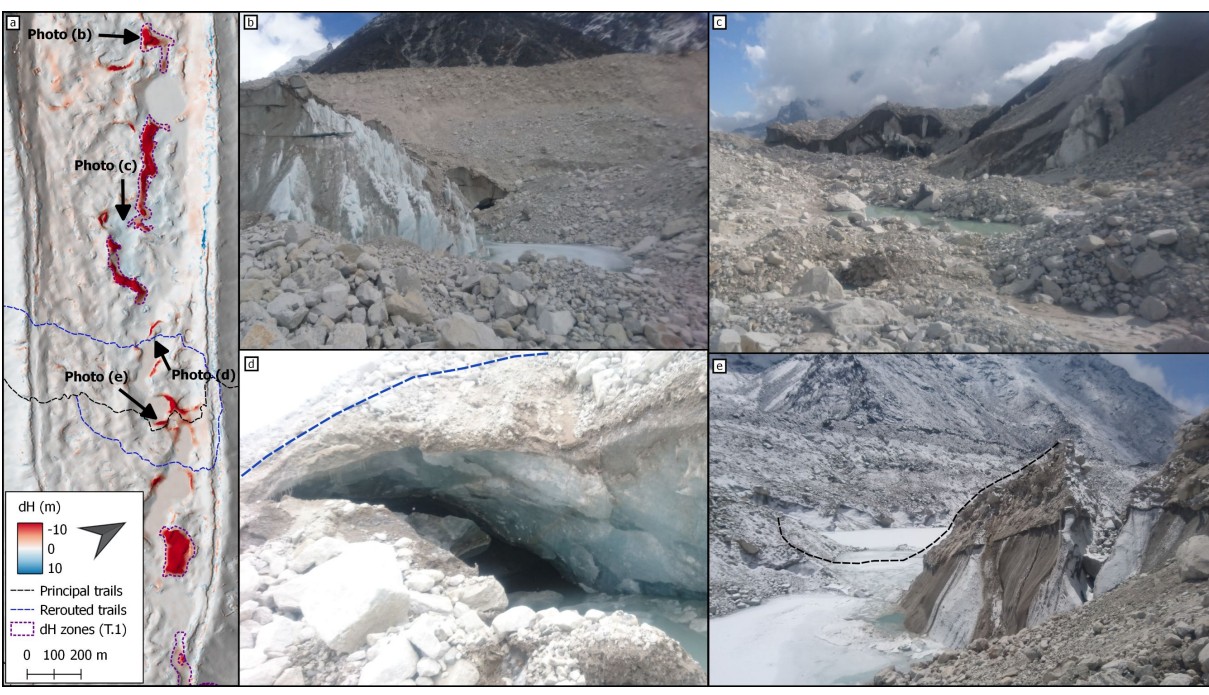

**Figure 5.** Surface changes on Khumbu Glacier. Surface lowering, rerouting of the Kongma La trail, and positions of select photos of enhanced change over the lowest three km of Khumbu Glacier (a). The area of a conduit collapse (Zone E), with visible water flowing towards the exposed conduit entrance (b). A zone of fluvially-reworked debris directly down-glacier from the conduit collapse in Zone F, and leading to exposed shallow conduits in the background (c). A cavernous englacial conduit exposure directly beneath the rerouted Kongma La trail (d). The route of the pre-event Kongma La trail, now cut off by a fresh conduit collapse (e, at right) and coalescing ponds. Photos from May 2018.



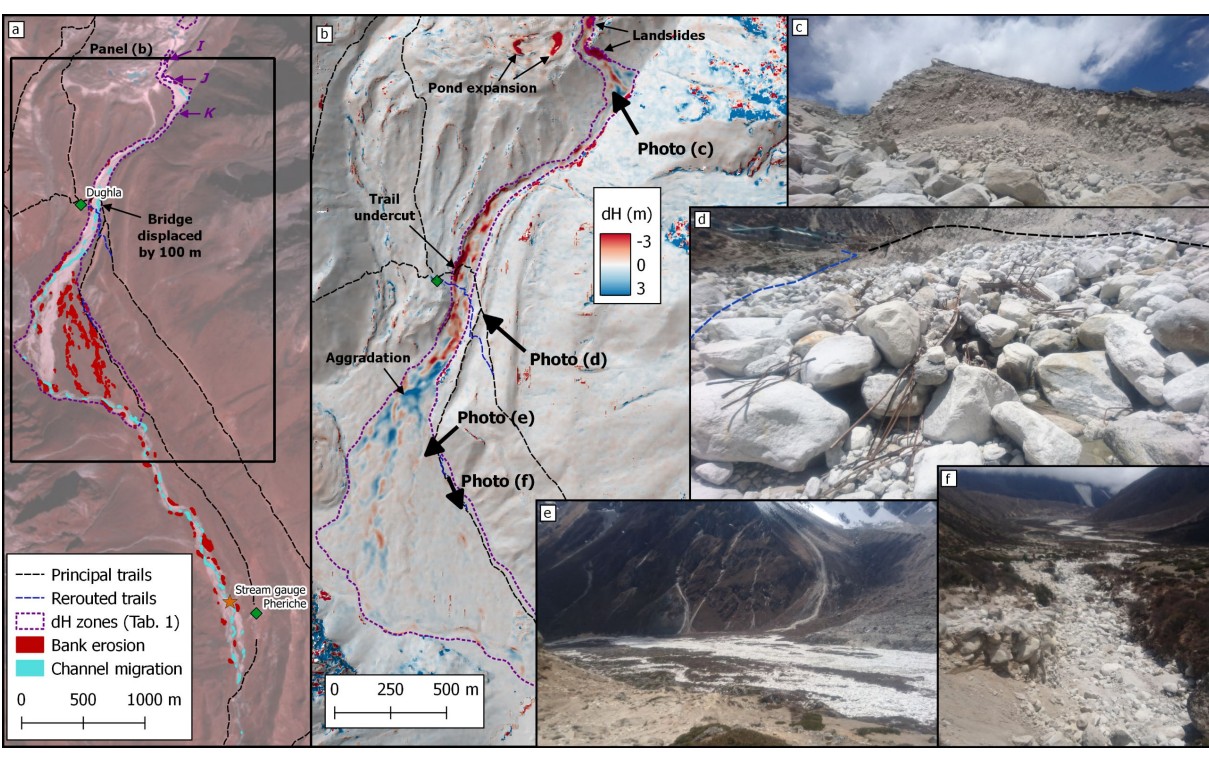

**Figure 6.** Geomorphic effects of the outburst flood below Khumbu Glacier showing extensive changes in vegetation cover due to bank erosion and migration of the stream channel 4 km downstream to Pheriche (a). Surface lowering associated with fluvial erosion and aggradation in the Khumbu proglacial system, and locations for select photos (b). A fresh landslide scarp (Zone J) directly below the Khumbu outlet (c). Remnants of a pedestrian bridge destroyed, carried 100 m downstream, and buried by the outburst, also indicating route of the trail before and after the outburst, with Dughla in the background (d). The Khumbu outwash plain in May 2018, showing widespread fluvially-reworked debris (e). A secondary channel used by the outburst flood, leading to > 1 m incision (f).




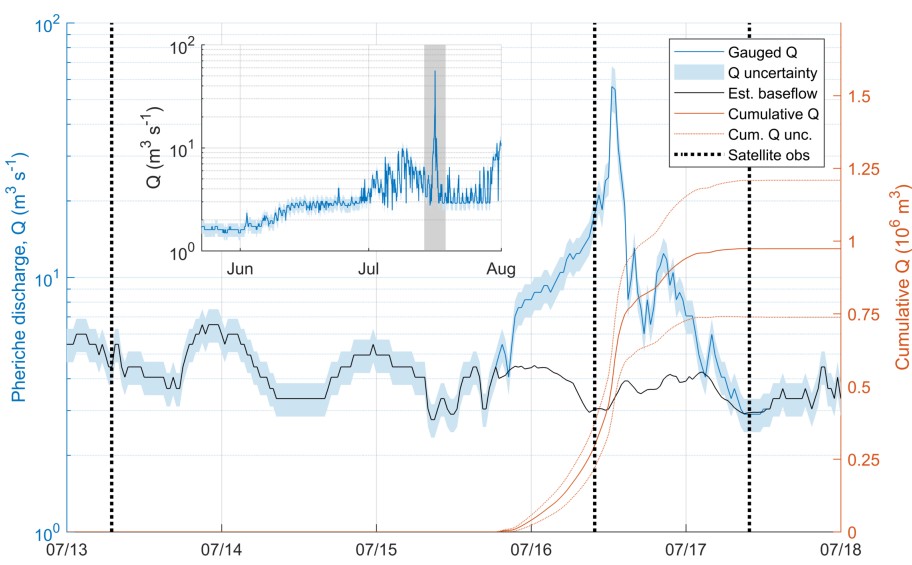

**Figure 7.** Pheriche discharge record during the outburst flood and cumulative flood volume, also indicating timing of PlanetScope observations. Inset shows the discharge record throughout the 2017 monsoon. Note the log scale for discharge.

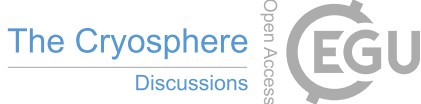



**Table 1.** Measured elevation changes associated with the lake drainage and outburst within key zones identified with the Pléiades stereo-imagery. CS and Kh denote Changri Shar and Khumbu Glaciers, respectively. Zones are identified in Figure 3. $\Delta$ V expresses the total volumetric change in each zone, and the full uncertainty based on $\sigma_{dH}$ for the zone area. 'V added' and 'V removed' are based on the elevation changes exceeding $\sigma_{dH}$. $\overline{dH}$ is the mean change in elevation within the zone, with uncertainty $\sigma_{dH}$.

| Zone | Description | Area (m$^2$) | $\Delta$ V (m$^3$) | V added (m$^3$) | V removed (m$^3$) | $\overline{dH}$ (m) |
|------|-------------|-------------|----------|-----------|-------------|----------|
| A | Emergence at CS terminus | 10,020 | -107,050±6,380 | 0 | -107,050 | -10.7±0.6 |
| B | Landslide and erosion in CS proglacial gorge | 16,030 | -186,030±9,780 | 50 | -186,080 | -11.6±0.6 |
| C | Surface lowering at Kh entrance | 27,870 | -185,970±16,130 | 400 | -186,330 | -6.7±0.6 |
| D | Sediment deposition at Kh entrance | 8,560 | 32,710±5,530 | 32,900 | -300 | 3.8±0.6 |
| E | Kh conduit collapse 1 | 9,400 | -49,090±6,030 | 150 | -49,110 | -5.2±0.6 |
| F | Kh conduit collapse 2 | 18,770 | -149,660±11,290 | 130 | -149,670 | -8.0±0.6 |
| G | Kh conduit collapse 3 | 9,900 | -88,130±6,320 | 40 | -88,130 | -8.9±0.6 |
| H | Kh conduit collapse 4 | 16,820 | -167,050±10,220 | 20 | -167,010 | -9.9±0.6 |
| I | Landslide 1 at Kh outlet | 670 | -4,200±530 | 0 | -4,200 | -6.3±0.6 |
| J | Landslide 2 at Kh outlet | 2,860 | -21,280±2030 | 0 | -21,270 | -7.4±0.6 |
| K | Kh outwash plain and proglacial channel | 831,830 | -80,210±320,080 | 112,860 | -180,890 | -0.10±0.4 |



**Table 2.** Areal changes along the Khumbu proglacial stream preceding (2012-2015) and encompassing (2016-2017) the lake outburst. Channel migration refers to the change in wetted area determined by NDWI thresholding, and bank erosion corresponds to the removal of vegetation in the channel area, identified by large NDVI differences.

| Distance from Khumbu outlet (km) | Area of channel migration (m$^2$) | | Area of bank erosion (m$^2$) | |
|---|---|---|---|---|
| | 2012-2015 | 2016-2017 | 2012-2015 | 2016-2017 |
| 1 | 0 | 1825 | 0 | 275 |
| 2 | 0 | 8325 | 0 | 5500 |
| 3 | 6225 | 4000 | 0 | 66800 |
| 4 | 3300 | 7050 | 0 | 10650 |
| 5 | 1175 | 6600 | 0 | 17475 |
| 6 | 4700 | 4925 | 1300 | 4000 |
| 7 | 3775 | 2475 | 1600 | 2100 |
| 8 | 5125 | 7725 | 425 | 6125 |
| 9 | 900 | 6150 | 2250 | 2425 |
| 10 | 0 | 0 | 350 | 0 |
| 11 | 7600 | 3625 | 200 | 1850 |
| Total | 32800 | 52700 | 6125 | 117200 |
| Change threshold | $\geq 0.081$ | $\geq 0.083$ | $\leq -0.160$ | $\leq -0.185$ |

*Author contributions.* ESM and DJQ planned the study. ESM analysed the supraglacial lake area timeseries, analysed glacial and proglacial elevation changes, and led the manuscript writing. CSW analysed proglacial stream migration and bank erosion. FB and EB processed the Pléiades tri-stereo imagery and coregistered DEMs and orthoimagery. ME installed, calibrated and collected data from the Pheriche stream gauge. ESM, KEM, and DJQ conducted fieldwork to assess and interpret geomorphic changes. All authors contributed to the interpretation of changes and manuscript preparation.

*Competing interests.* The authors declare that they have no conflict of interest.

*Acknowledgements.* The authors gratefully acknowledge Planet for provision of imagery. This research was supported by the 'EverDrill' Natural Environment Research Council Grant awarded to the University of Leeds (NE/P00265X) and Aberystwyth University (NE/P002021). CSW acknowledges support from the NASA High Mountain Asia grant NNX16AQ62G. EB acknowledges support from the French Space Agency (CNES) through the TOSCA program. The discharge data were collected as part of two projects funded by the French National Research Agency (ANR), France (references ANR-09-CEP-0005-04/PAPRIKA and ANR-13-SENV-0005-04/05-PRESHINE). KEM is funded





by an AberDoc PhD Studentship. ESM, KEM, and DJQ acknowledge Himalayan Research Expeditions and Mahesh Magar in particular for field logistical support. We are thankful for constructive discussions with M Kirkbride.





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
