# Peer review of "Glacial and geomorphic effects of a supraglacial lake drainage and outburst event, Everest region, Nepal Himalaya"

_The Cryosphere, 2018_

## Referee Comment (RC1) · D. Benn (Referee) · 23 Aug 2018

This is an excellent paper, which provides very rare detailed documentation of a transient drainage event from a Himalayan glacier. It is exceptionally well written, and presents the methods and results in a clear and logical way. The interpretations and conclusions are sound and convincing. I can find no fault with the paper, and have only a few remarks, mostly relating to pertinent unpublished observations.

p. 2, line 20: nature of the flow path into Khumbu Glacier. In 2006, Jason Gulley and I entered an ice cave in the margin of Khumbu Glacier at the bottom of the Changri proglacial gorge. The entrance led into a low, wide passage that trended along parallel to the slope. The passage was floored by boulders resting on bedrock, so any water entering the glacier would flow along the ice-bed interface, at least initially. This was also one of the most dangerous places we had ever been, owing to rocks occasionally bouncing down the gorge, so we did not linger long enough to make any surveys.

[Figure]

Perhaps the fact that the system was subglacial in 2006 could be added as a 'pers. comm.'

This observation also helps to support the authors' interpretation of the flood flowpath through Khumbu Glacier, presented on p. 9, line 15. Interestingly, there is also evidence of a sub-marginal / englacial drainage system on Ngozumpa Glacier, which is intermittently connected to the supraglacial / englacial system. The same may be true on Khumbu Glacier - although the flood likely bypassed the supraglacial - englacial system inferred by Irvine-Fynn et al. (2017), the two systems may not be entirely and perennially separate.

p. 7, line 7: hydrofracture is unlikely because the lower Changri glacier is stagnant - see Benn et al. 2009 for a discussion of the conditions required for hydrofracture on Khumbu Glacier.

---

## Referee Comment (RC2) · D. Rounce (Referee) · 11 Sep 2018

**Review of "Ablative and geomorphic effects of a supraglacial lake drainage and outburst event, Nepal Himalaya"**
**by Miles et al.**

General Comments

This study uses a series of high-resolution satellite images to document the filling and coalescing of supraglacial ponds on Changri Shar Glacier that rapidly drained in July 2017 causing a glacier outburst flood. High-resolution DEMs were used to analyze the geomorphic changes caused by the outburst flood, and the social impacts were considered as well. The study was very well written and easy to follow. Figures, albeit a little small at times, contained significant amounts of information that supported the text well. The study's use of high-resolution satellite imagery combined with multiple DEMs enabled a very novel approach for quantifying the flood volume. The flow measurements and field observations also provided unique insight into the timing and path of the flood, and supported the interpretation of events. The discussion contextualized the study well, highlighting (i) the power of being able to use a suite of remote sensing products to observe and quantify glacier outburst floods, and (ii) the impacts that these glacier outburst floods can have on local communities. Given the lack of observations of glacier outburst floods, the holistic nature and level of detail in which this event was analyzed, and how well written this study was, I recommend this manuscript be accepted. A few minor comments may be found below.

Minor Comments

Figure 1 – There is a lot of information in this figure, but I found key aspects of the figure a bit difficult to read. For example, this figure introduces readers to the general area, so the names of the glaciers should be clear (they are very small and hard to read). After looking at the figure for a while, the inset figure clearly shows the maximum area, but the legend does not state this nor is this mentioned in the caption. I would simply make note of this in the caption, so the reviewer knows they are looking at how the maximum lake extent fills and drains. If possible make the text larger.

P3 L23-28 – Does this mean you avoided all areas that had a supraglacial pond or ice cliff in the previous year? Please add a sentence here detailing how you identified areas where surface lowering was not attributable to cliffs and ponds, since it is not very clear.

P4 L9 – "by *a* cloud" or "by *clouds*"?

P5 L16 vs. L17 & L32 – I think it is better to be explicit when referring to the zones like L16 "Zone A in Figure 2"; however, on L17 and 32, the zones are just stated. I suggest being consistent throughout the text in how you refer to them. Either always refer to them as Zone _ in Figure 2, or change L16.

P7 L9 – This appears to reference Figure 4c, not 3c.

P7 L24 – This appears to reference Rounce et al. (2017) not Rounce et al. (2016).

P7 L26-28 – I found this sentence unclear and difficult to read.  What do you mean by "of this area in similar conditions"?  Also, "examining available historic satellite image archives we have not found" does not make sense – perhaps split this into two sentences: "The area of bank erosion is greatly magnified during 2016-17.  This magnitude of geomorphic change appears to be uncommon, since we were unable to find similar areas of bank erosion in any of the historic satellite image archives"?

P7 L29 – It appears that at least a portion of the second peak is simply due to the diurnal signal caused by the melting of the glacier.  On July 14$^{th}$, the flow increased by approximately 3 m$^3$ s$^{-1}$, compared to this second peak where it increases around 3.5 – 4 m$^3$ s$^{-1}$; hence, it doesn't seem unreasonable that this is simply the extra discharge coming from the glacier melt.  It's timing is consistent as well.  This seems much more likely than a possible blockage, since one would expect that the flood would generate very efficient channels, which would make something getting blocked unlikely.

P8 L25 – The use of "low" here is a bit awkward.  Consider "melt-inhibiting thick debris *near the terminus* on such glaciers" or something similar.

P9 L2 – "region" not "regional"

---

## Author Response (AR1)

**Glacial and geomorphic effects of a supraglacial lake drainage and outburst event, Everest region, Nepal Himalaya**

Evan S. Miles, C. Scott Watson, Fanny Brun, Etienne Berthier, Michel Esteves, Duncan J. Quincey, Katie E. Miles, Bryn Hubbard, and Patrick Wagnon

Final Response

12/11/2018

Dear Professor Farinotti,

We have received two reviews to our manuscript, both of which were extremely supportive of our analysis and interpretations, and we have adapted the manuscript in light of their constructive comments. In the document below, the reviewers' comments appear in italicized blue, followed by our response from the open discussion in normal black text, and in red font we have indicated the changes made to the updated manuscript. Along with this letter, we are submitting the revised version of the manuscript and a version with changes marked.

In addition, we have made very minor textual changes for clarity and adjusted the precision of values in Table 1 to better reflect our confidence in the measurements.

There are a two minor changes to the manuscript that were not directly stimulated by the reviewers, and which I would like to note. First, based on internal discussions during the review process, we decided to revise the title to 'Glacial and geomorphic...' as the changes at the glacier surfaces are not necessarily strictly indicative of ablation. Second, and also after internal discussion during the review process, we have decided to include Bryn Hubbard as a co-author. This is due to his support of our field investigations, and to acknowledge extensive discussions between the authors and Bryn related to the drainage pathways of these glaciers, particularly in the situation of an outburst flood. Those discussions led, for example, to the development of a third hypothesis for the double-peak flood hydrograph, which is now included in the text.

Thank you for your consideration of our revised manuscript, which we hope is now acceptable for publication. Please address correspondence to me at evan.miles@wsl.ch

Sincerely,

Evan Miles and Co-authors

for D Min

**Comments by Reviewer #1, Doug Benn, and responses:**

This is an excellent paper, which provides very rare detailed documentation of a transient drainage event from a Himalayan glacier. It is exceptionally well written, and presents the methods and results in a clear and logical way. The interpretations and conclusions are sound and convincing. I can find no fault with the paper, and have only a few remarks, mostly relating to pertinent unpublished observations.

**Dear Doug,**

Thank you very much for your supportive review. We are very happy that you enjoyed the paper, and are of course delighted to hear of the independent evidence supporting our interpretations regarding the flood's flowpath. We respond to your specific comments individually below, with your comments appearing in italics and our response in normal text.

**Kind regards,**

**Evan and Co-authors**

p. 2, line 20: nature of the flow path into Khumbu Glacier. In 2006, Jason Gulley and I entered an ice cave in the margin of Khumbu Glacier at the bottom of the Changri proglacial gorge. The entrance led into a low, wide passage that trended along parallel to the slope. The passage was floored by boulders resting on bedrock, so any water entering the glacier would flow along the ice-bed interface, at least initially. This was also one of the most dangerous places we had ever been, owing to rocks occasionally bouncing down the gorge, so we did not linger long enough to make any surveys.

Perhaps the fact that the system was subglacial in 2006 could be added as a 'pers. comm.'

This observation also helps to support the authors' interpretation of the flood flowpath through Khumbu Glacier, presented on p. 9, line 15. Interestingly, there is also evidence of a sub-marginal / englacial drainage system on Ngozumpa Glacier, which is intermittently connected to the supraglacial / englacial system. The same may be true on Khumbu Glacier - although the flood likely bypassed the supraglacial englacial system inferred by Irvine-Fynn et al. (2017), the two systems may not be entirely and perennially separate.

P2, L20: The unpublished observations from 2006 are very interesting indeed, and we will include a citation to these observations as suggested. Several of the authors have passed through the gorge to reach Gorak Shep from the glacier surface, but did not venture into the cave and inlet. We heartily agree that this is a very dangerous area, especially during the melt season. Nonetheless the observation of the passage's floor characterized by boulders resting on bedrock is extremely valuable to confirm our interpretation of a subglacial flowpath from this point (at least initially). Thank you for sharing your observations.

Text modified to read: "The stream has cut into the lateral margin of Khumbu Glacier, leading to development of a large bare ice cliff. From this position, water initially flows into a low, wide passage along the ice-bed interface (D. Benn, pers. comm., 23 August 2018)."

It is interesting as well to speculate whether this flowpath simply connects to a sub-marginal drainage system from up-glacier, or whether it is the sole cause for this particular flowpath. Our discussions of the structure of the Khumbu drainage system are still ongoing, but in May 2017 we were able to trace a

surface channel to just up-glacier of the Changri gorge. It is still unclear whether these flowpaths (the surface/near-surface from the glacier's upper debris area and the lateral input from Changri) connect directly or closer to the terminus, or the nature of their connection with the supraglacial-englacial system.

As you suggest, the systems may not be entirely and perennially separate, and we may have presented the 'bypass' too simply. One very possible scenario is that the flood's passage through Khumbu Glacier overpressured the subglacial system, leading to the partial emergence of the flood at the surface flowing through the linked ponds and contributing to the double (or triple) peak in the hydrograph. We will discuss this possibility briefly in the revised manuscript, but what is very clear is that the flood exploited a subglacial flowpath for part of its transit through Khumbu Glacier.

Discussion of the multiple peaks now reads: "The double peak of discharge observed at Pheriche (Figure 7) is unusual for outburst floods. A possible cause is the blockage of the Khumbu stream inlet by the landslide in the Changri Shar proglacial gorge (Figure 4). This is likely to have initiated around peak flow through the gorge, and could have led to a substantial decline in discharge, followed by a later, sudden increase as preferential flowpaths developed through the debris (Gulley et al., 2009b). A second explanation is the possibility of multiple flowpaths for the flood through the lower part of Khumbu Glacier. As Khumbu Glacier exhibits a low terminus slope and high hydraulic base level, the flood may have temporarily overwhelmed the subsurface drainage network and, exploiting fractures and secondary pathways common for these glaciers, partially emerged at the glacier surface. This would result in two or more flowpaths of differing efficiency, possibly leading to distinct discharge peaks on the Pheriche hydrograph. This possibility is supported by the appearance of highly turbid water in the ponds between zones E-G (Figure 2) during drainage. A third possibility is that the increased discharge late on 16 July corresponds to delivery of water stored elsewhere within the glacier system. Such stored water might connect to the drainage system more efficiently by the opening of conduits and channels during the flood. Regardless, 5 it is clear that the increase in discharge at Pheriche only lasts until 10:00 on 17 July, so the flood's direct contribution to discharge was short-lived."

*p.* 7, line 7: hydrofracture is unlikely because the lower Changri glacier is stagnant - see Benn et al. 2009 for a discussion of the conditions required for hydrofracture on Khumbu Glacier.

**P7, L7: True, we will adjust this text in the revision.**

Modified text reads: "Based on the lack of down-glacier surface change on Changri Shar, the lake must have drained englacially or subglacially, rather than along the surface. Hydrofracture is an unlikely scenario as the ice is nearly stagnant in this area; rather, this could have been accomplished by penetrating the internal blockage or establishing a new connection to relict conduits."

**Comments by Reviewer #2, Dave Rounce, and responses:**

This study uses a series of high-resolution satellite images to document the filling and coalescing of supraglacial ponds on Changri Shar Glacier that rapidly drained in July 2017 causing a glacier outburst flood. High-resolution DEMs were used to analyze the geomorphic changes caused by the outburst flood, and the social impacts were considered as well. The study was very well written and easy to follow. Figures, albeit a little small at times, contained significant amounts of information that supported the text well. The study's use of highresolution satellite imagery combined with multiple DEMs enabled a very novel approach for quantifying the flood volume. The flow measurements and field observations also provided unique insight into the timing and path of the flood, and supported the interpretation of events. The discussion contextualized the study well, highlighting (i) the power of being able to use a suite of remote sensing products to observe and quantify glacier outburst floods, and (ii) the impacts that these glacier outburst floods can have on local communities. Given the lack of observations of glacier outburst floods, the holistic nature and level of detail in which this event was analyzed, and how well written this study was, I recommend this manuscript be accepted. A few minor comments may be found below.

**Dear Dave,**

Thank you for your careful and positive review. Thank you also for the comments and questions, which will improve the manuscript's clarity. We will certainly reconsider the figure size (and especially the font size within the figures) to improve their readability for the revised manuscript. We respond to your specific comments individually below, with your comments appearing in blue italics and our response in normal text.

Kind regards,

**Evan and Co-authors**

Figure 1 – There is a lot of information in this figure, but I found key aspects of the figure a bit difficult to read. For example, this figure introduces readers to the general area, so the names of the glaciers should be clear (they are very small and hard to read). After looking at the figure for a while, the inset figure clearly shows the maximum area, but the legend does not state this nor is this mentioned in the caption. I would simply make note of this in the caption, so the reviewer knows they are looking at how the maximum lake extent fills and drains. If possible make the text larger.

Thank you for the suggestions. We will certainly increase the font size in this figure (and others) for key aspects, and will include a reference in the caption to the display of maximum lake area in the inset figures.

We have reexamined the font sizes for all labels in the figure and have made numerous adjustments to improve the clarity of the figure.

P3 L23-28 – Does this mean you avoided all areas that had a supraglacial pond or ice cliff in the previous year? Please add a sentence here detailing how you identified areas where surface lowering was not attributable to cliffs and ponds, since it is not very clear.

Regarding this section, we agree that this text was slightly ambiguous, and should be clarified. The current text reads 'not solely attributable to ice cliffs and supraglacial ponds' and later mentions 'clearly

associated with the lake drainage.' By this we do not mean that cliffs and ponds were entirely excluded or played no role, but that we could not explain the zone of surface lowering only by the presence of cliffs and ponds. Our rationale appears below, but we will carefully modify the text to succinctly describe that we have avoided pond water-level lowering and areas that had a thin, arcuate form, but focused specifically on broad areas of enhanced elevation change with initial changes visible in the Planet imagery.

The delineation process for zones of enhanced change focused on identifying zones of considerable elevation change with three key characteristics:

- 1) We first ensured that zones of elevation change were not attributable to pond water level change. This is straightforward to avoid as such zones would have been ponds in the March Pleiades image.
- 2) We aimed to identify zones inexplicable by ice cliff backwasting. Backwasting rates for Khumbu Glacier are 1-6 cm d-1 (Watson et al, 2017), depending on season and local characteristics. Over the period between our two Pleiades DEMs (266 days between 23 March and 14 December 2017), this would total 2.7-16.0 m of cliff retreat. This linear change could be adjusted by the cliff's advection down-glacier (e.g. Brun et al, 2018), but Khumbu is nearly stagnant below the Changri inlet, so we neglect this. As cliffs tend to have arcuate or linear forms several 10s of meters in length, but backwaste up to ~10 m during the melt season, the features leave a characteristic thin arc of enhanced mass loss in our dH data, which is due to the high spatial resolution of the DEM and the relatively short interval between acquisitions. We ignored these forms (clearly visible in Figure 2) entirely, but focused on broad (i.e. >40 m across), continuous areas of elevation change.
- 3) The third necessary characteristic is that minimal change was evident in the Planet imagery prior to the lake drainage.

Although this process was subjective, we were as conservative as possible. For example, field evidence suggested that many of the changes in the area shown in Figure 5e (location shown in 5a) were probably due to the passage of the flood, but the pattern of dH in this area appears similar to ice cliff backwasting, so we did not include it in our analysis. Without a doubt, it is not possible to entirely separate the effects of cliff and flood: the passage of water directly leads to exposure of steep, bare ice (i.e. a cliff).

Revised text now reads "This geodetic difference encompassed the majority of the ablation season, so for the glaciers we focused on zones of enhanced surface lowering not solely attributable to ice cliffs and supraglacial ponds, which are known hot spots of melt for Himalayan debris covered glaciers (e.g. Sakai et al., 2002). Ice cliffs tend to have curvilinear forms, with their planimetric length much greater than their width (e.g. Brun et al., 2016; Kraaijenbrink et al., 2016). For our study area, we are able to neglect advection and emergence of these features due to glacier dynamics (e.g. Brun et al., 2018), as the lowest 5 km of Khumbu Glacier is stagnant (Rounce et al., 2018). Over a short interval, melt along the inclined cliff surface was thus expressed as a thin arc of surface lowering (e.g. Immerzeel et al., 2014), clearly identifiable in Figure 2.We ignored these cliff areas and areas of elevation change within ponds. We thus identified 11 zones of prominent elevation change that were clearly associated with

the lake drainage according to the PlanetScope and RapidEye imagery (Table 1, Figure 2). Field visits in May 2017, October 2017, and May 2018 enabled direct observation of many of the most prominent zones of change."

P4 L9 – "by a cloud" or "by clouds"?

True, "by clouds." We will adjust this in the revised manuscript.

Adjusted.

P5 L16 vs. L17 & L32 – I think it is better to be explicit when referring to the zones like L16 "Zone A in Figure 2"; however, on L17 and 32, the zones are just stated. I suggest being consistent throughout the text in how you refer to them. Either always refer to them as Zone \_ in Figure 2, or change L16.

Thank you for the suggestion. We had decided to refer the reader to the pertinent table and figure at the first instance only, but we agree that it is probably easier for the reader if we refer them at each instance.

Adjusted.

P7 L9 – This appears to reference Figure 4c, not 3c.

Thank you! We will adjust this in the revised manuscript.

Corrected.

P7 L24 – This appears to reference Rounce et al. (2017) not Rounce et al. (2016).

You are correct, our apologies! This will be corrected in the revised manuscript.

**Corrected.**

P7 L26-28 – I found this sentence unclear and difficult to read. What do you mean by "of this area in similar conditions"? Also, "examining available historic satellite image archives we have not found" does not make sense – perhaps split this into two sentences: "The area of bank erosion is greatly magnified during 2016-17. This magnitude of geomorphic change appears to be uncommon, since we were unable to find similar areas of bank erosion in any of the historic satellite image archives"?

Thank you for the suggestion, which we will implement in the revised manuscript.

The text now reads: "The 2016-2017 NDWI and NDVI changes show a greater magnitude of channel migration despite the shorter interval. The area of bank erosion is also greatly enhanced during 2016-2017. The magnitude of geomorphic change associated with the flood appears to be uncommon, since we were unable to find similar areas of bank erosion in any of the historic satellite image archives."

P7 L29 - It appears that at least a portion of the second peak is simply due to the diurnal signal caused by the melting of the glacier. On July 14th, the flow increased by approximately 3 m3 s -1, compared to this second peak where it increases around 3.5 - 4 m3 s -1; hence, it doesn't seem unreasonable that this is simply the extra discharge coming from the glacier melt. It's timing is consistent as well. This seems much more likely than a possible blockage, since one would expect that the flood would generate very efficient channels, which would make something getting blocked unlikely. We certainly agree that some of the second peak is simply due to the diurnal signal, and have estimated that portion based on the diurnal discharge patterns preceding and following the event (the black line in Figure 7). The second peak corresponds to an increase of 6.5 m3 s-1 (see P6 L14-18; note that Figure 7 has a logarithmic scale), which is greater than the diurnal variation preceding the event.

The blockage hypothesis corresponds to the landslide in the Changri gorge, which definitely occurred on the 16th. This deposit of mass would have choked the entrance to the sub-marginal drainage path, preventing access to the englacial and subglacial channels altogether, but the debris blockage would be unlikely to prevent drainage for long, and would thus be a potential candidate for one of the peaks later on the 16th.

A third possible explanation, that of multiple flowpaths, has arisen from internal discussions since the manuscript submission, and we will also adapt the manuscript to briefly include it in the discussion. We think it likely that the flood's passage through Khumbu Glacier would have temporarily overpressured the subglacial system. In this case, water would try to exploit weaknesses in the ice to drain to the surface. We see evidence for surface routing of at least part of the flood from the zones of enhanced elevation change (e.g. Figure 5) and from increased turbidity of the chain of terminal ponds on the 16th and 17th of July. However, this does not mean that the entire flood would have been routed to the surface; instead, only the water which could not be accommodated by subglacial and englacial conduits would find its way to the surface. As the surface flowpath is inefficient (Irvine-Fynn et al, 2017), this would lead to at least two distinct traces at the Pheriche station.

We will be sure to represent all three potential hypotheses in the revised manuscript.

The revised text now reads: "The double peak of discharge observed at Pheriche (Figure 7) is unusual for outburst floods. A possible cause is the blockage of the Khumbu stream inlet by the landslide in the Changri Shar proglacial gorge (Figure 4). This is likely to have initiated around peak flow through the gorge, and could have led to a substantial decline in discharge, followed by a later, sudden increase as preferential flowpaths developed through the debris (Gulley et al., 2009b). A second explanation is the possibility of multiple flowpaths for the flood through the lower part of Khumbu Glacier. As Khumbu Glacier exhibits a low terminus slope and high hydraulic base level, the flood may have temporarily overwhelmed the subsurface drainage network and, exploiting fractures and secondary pathways common for these glaciers, partially emerged at the glacier surface. This would result in two or more flowpaths of differing efficiency, possibly leading to distinct discharge peaks on the Pheriche hydrograph. This possibility is supported by the appearance of highly turbid water in the ponds between zones E-G (Figure 2) during drainage. A third possibility is that the increased discharge late on 16 July corresponds to delivery of water stored elsewhere within the glacier system. Such stored water might connect to the drainage system more efficiently by the opening of conduits and channels during the flood. Regardless, 5 it is clear that the increase in discharge at Pheriche only lasts until 10:00 on 17 July, so the flood's direct contribution to discharge was short-lived."

*P8 L25 – The use of "low" here is a bit awkward. Consider "melt-inhibiting thick debris near the terminus on such glaciers" or something similar.*

Agreed, thank you.

Adjusted.

**P9 L2 – "region" not "regional"**

Agreed, thank you.

Adjusted.

**References in the response**

- Brun, F., Wagnon, P., Berthier, E., Shea, J. M., Immerzeel, W. W., Kraaijenbrink, P. D. A., Vincent, C., Reverchon, C., Shresta, D., and Arnaud, Y. (in review, 2018). Can ice-cliffs explain the debriscover anomaly? New insights from Changri Nup Glacier, Nepal, Central Himalaya, *The Cryosphere Discuss.*, https://doi.org/10.5194/tc-2018-38.
- Watson, C. S., Quincey, D. J., Smith, M. W., Carrivick, J. L., Rowan, A. V., & James, M. R. (2017). Quantifying ice cliff evolution with multi-temporal point clouds on the debris-covered Khumbu Glacier, Nepal. *Journal of Glaciology*, *63*(241), 823–837. https://doi.org/10.1017/jog.2017.47

**Ablative Glacial** and geomorphic effects of a supraglacial lake drainage and outburst event, **Everest region**, Nepal Himalaya**

Evan S. Miles1,2, C. Scott Watson1,3, Fanny Brun4,5, Etienne Berthier4, Michel Esteves5, Duncan J. Quincey1, Katie E. Miles6, Bryn Hubbard6, and Patrick Wagnon5 1School 
[revised manuscript text omitted]